# Multi-Attribute Decision-Making Approach for a Cost-Effective and Sustainable Energy System Considering Weight Assignment Analysis

Keifa Vamba Konneh [1,*] , Hasan Masrur [1] , Mohammad Lutfi Othman [2] , Hiroshi Takahashi [3],
Narayanan Krishna [4] and Tomonobu Senjyu [1,*]

1 Department of Electrical & Electronics Engineering, Faculty of Engineering, University of the Ryukyus,
  1 Senbaru, Nishihara-cho, Nakagami, Okinawa 903-0213, Japan; k198676@eve.u-ryukyu.ac.jp
2 Advanced Lightning and Power Energy Research (ALPER), Department of Electrical and Electronic
  Engineering, Faculty of Engineering, University Putra Malaysia (UPM), Serdang 43400, Malaysia;
  lutfi@upm.edu.my
3 Fuji Electric Co., Ltd., Tokyo 141-0032, Japan; takahashi-hirosi@fujielectric.com
4 Department of Electrical and Electronic Engineering, SASTRA Deemed University, Thanjavur 613401, India;
  narayanan@eee.sastra.edu
* Correspondence: k208677@eve.u-ryukyu.ac.jp (K.V.K.); b985542@tec.u-ryukyu.ac.jp (T.S.)

**Abstract:** The need for inexpensive and sustainable electricity has become an exciting adventure due to the recent rise in the local population and the number of visitors visiting the Banana Islands. Banana Islands is a grid-isolated environment with abundant renewable energy, establishing a hybrid renewable energy-based power system may be a viable solution to the high cost of diesel fuel. This paper describes a dual-flow optimization method for electrifying the Banana Islands, a remote island in Sierra Leone. The study weighs the pros and cons of maintaining the current diesel-based power setup versus introducing a hybrid renewable energy system that takes backup component analysis into account. Hybrid Optimization of Multiple Energy Resources (HOMER) software is used in the first optimization to optimally design the various system configurations based on techno-economic and environmental characteristics. A Multi-Attribute Decision-Making (MADM) Model that takes into account in the second optimization, the Combinative Distance-based Assessment System (CODAS) algorithm, and various methods of assigning weights to the attributes is used to rank the best configuration. The results show that the hybrid renewable energy system is a better option for electrifying the Banana Islands than the current stand-alone system. The Analytical Hierarchy Process (AHP) method of weight assignment was found to be superior to the Entropy method. Biogas generator-assisted hybrid configurations outperformed diesel generator-assisted hybrid configurations. With an optimum design of 101 kW PV, 1 wind turbine, 50 kW biogas, 86 batteries, and a 37.8 kW converter, the PV-wind-biogas-battery system is rated as the best configuration. It has a net present cost (NPC) of $487,247, a cost of energy (COE) of $0.211/kWh, and $CO_2$ emission of 17.5 kg/year. Sensitivity analyses reveal that changes in the rate of inflation and the cost of storage have a significant effect on the overall cost of the configuration.

**Keywords:** hybrid systems; techno-economic-environmental analysis; off-grid; multi-attributes decision-making; weight assignment

## 1. Introduction

Power system engineers have found that designing and preparing the configuration of power systems for grid-isolated settlements has become a major bottleneck. Obtaining a cost-effective and long-lasting power configuration will help to boost economic growth. This is much more difficult in settlements that do not have access to electricity, as the authors in [1] demonstrate. According to [2], about 55% of Africa's rural population does not have access to electricity. Sierra Leone, like many other developing countries, has been

adopting recent energy policies to electrify rural communities [3]. The authors of [4] listed a number of approaches focused on a collection of indicators that the government might use to incorporate hybrid mini-grid operations for rural electrification. Sierra Leone is one of the best countries in West Africa for tourism, with many coastal settlements, according to [5]. A large exodus of tourists visits these coastal settlements, but most of their activities are distorted by poor electricity access. Many of these settlements are located away from grid networks and are surrounded by dense jungles and rough terrain, which makes grid extension difficult. These communities rely on kerosene lamps and stand-alone diesel generators to provide electricity, according to [6]. The high operating costs of diesel generators, combined with serious environmental pollution, make it impossible to provide affordable, continuous, and sustainable electricity to rural communities, as cited in [7,8]. Alternatives to standalone diesel generators have arisen in the form of renewable energy power systems. Sierra Leone has enormous renewable energy potential in the form of biomass from agricultural wastes, hydro, wind, and solar, but little effort has been made to investigate these resources. According to [9], hydropower is the largest renewable energy potential in Sierra Leone with an estimated capacity of 5000 MW covering 300 sites nationwide. Average solar radiation ranges between 4.1–5.2 kWh/m$^3$/day. Wind speeds vary from 3–5 m/s, with gusts up to 8 m/s in mountainous areas. These renewable energy sources, however, cause major disruptions due to their intermittent existence, as demonstrated by [10–14]. As a result, hybrid renewable energy sources with complementary features must be built in order to sustain a stable or efficient power system. In the following literature [15–18], the techno-economic advantages of hybrid renewable energy systems are compared to diesel-based power systems.

Many studies have looked at diesel and biogas generators as backup components when other renewable energy sources, especially solar and wind resources, are unavailable [19–21]. In [22], the authors compared the feasibility of nine different system configurations for an off-grid system in southern Cameroon to improve sustainable power supply. The PV/diesel/small hydro/battery system was considered the optimum configuration with COE $0.443/kWh. The hybrid system was found to be a better choice for stability and reliability than a 100% renewable energy design, which is vulnerable to high uncertainties.. Kiflom et al. [23] optimized a cost-benefit analysis on a hybrid energy system to electrify a rural Ethiopian village. The hybrid PV-Wind-Diesel-Battery system had the lowest system cost and $CO_2$ emission of 37.3 tons/year as compared to the diesel only system. Ali et al. [24] evaluated various hybrid energy systems to supply electricity to a rural village in Iraq, taking into account techno-economic and environmental factors. The hybrid PV/hydro/diesel/battery system proved to be the most cost-effective and reliable choice for long-term electrification. Monowar et al. [25] assessed the efficiency of a hybrid energy system to determine its ability to provide power to a Malaysian resort. A variety of hybrid configurations were tested and compared to a standalone diesel generator. Results showed that the optimum hybrid configuration reduced costs by 18.5% and $CO_2$ by 52% compared to the diesel-only system, validating the supremacy of hybrid energy systems to a stand-alone diesel system. In Shibpur Campus, India, Tathagata et al. [26] used hybrid renewable energy sources to optimally size a smart microgrid. Since renewable energies have high intermittencies, it is often preferable to use other reliable sources of energy to ensure long-term access. The simulation results show that involving a biogas generator provided the necessary energy with no loss of power supply probability. Wei et al. [27] carried out an optimization process in South Khorasan, Iran, using a geographic information system module and a hybrid optimization algorithm to find the best location and equipment capacities. Simulation results confirmed that the hybrid PV-Diesel-Battery configuration reduced costs and greenhouse gas by 22.2% and 59.6%, respectively. In addition, the use of hybrid algorithm for the proposed framework was 14.1% more accurate compared to individual applied algorithms. Abhishek et al. [28] used a computational modeling method to determine the viability of a community hybrid energy system in two European cities: the United Kingdom and Bulgaria. Biomass was chosen because of its potential to provide

energy while also reducing household waste. While there was a significant difference in solar and wind availability between the two cities, the research found that biomass generators had the greatest share due to the vast reliability of the raw materials. The authors of Ref. [29] used four different optimization methods and three different battery technologies to perform a techno-economic study on an off-grid hybrid PV-biomass system in Egypt. The abundant biomass resource combined with the region's solar availability made the system optimum by ensuring a stable supply, according to the findings. In order to provide sustainable electricity, a feasibility study was conducted on a hybrid off-grid system in a remote location in Morocco. Various system configurations were studied in terms of technology and cost, and the PV-wind-biomass system was found to be the most effective. The results showed that biomass provided 48% of the electricity due to its consistent supply, resulting in the lowest greenhouse gas emissions [30]. Shakti et al. [31] proposed a hybrid PV-wind-biomass energy system for Patiala, Punjab, India. Comparative analysis of various algorithms was used in the simulations. The existence of a biogas generator, combined with abundant available resources, ensured that the load was fulfilled without constraint violations, according to a testing strategy that enabled one of the components, the wind turbine, to fail.

Though hybrid renewable energy configurations have proven to be more reliable and cost-effective than standalone diesel configurations, determining the best configuration can be difficult and time-consuming, particularly when there are several factors to consider (economic, technical, and environmental). Project planners are prone to prejudice, and they can choose the best configuration based on personal desires rather than sustainability. This necessitates the application of MADM. In [32], the authors conducted a state-of-the-art analysis of MADM strategies for making decisions in renewable energy systems applications. These methods, according to them, have been used to evaluate energy policies, choose the best renewable energy source for electricity generation, evaluate renewable energy sources, find the best location for a renewable energy plant, and choose the best energy alternatives. Our analysis will use the idea of choosing the best in this review. The assigning of weights to the considered attributes is one of the key bottlenecks in MADM techniques. Weights assignment has a significant effect on the decision result, so it must be given careful consideration when choosing a methodology. Subjective, objective, and integrated [33] are the three approaches for assigning weights. Where several parameters or characteristics are considered, MADM techniques have been used in a number of studies to choose the best energy system alternative. The authors of [34] used General Algebraic Modeling System (GAMS) software to optimize the different scenarios before ranking them using the ELimination Et Choix Traduisant la REalité (ELECTRE) MADM process. In Cameroon, Benyon et al. [35] conducted a sustainable energy planning study using a combination of the AHP and the Vlsekriterijumska Optimizacija I KOmpromisno Resenje system (VIKOR) to find the best hybrid technology combination. The authors of [36] developed a two-stage MADM analysis method for city-integrated hybrid mini-grid architecture. In the first step, HOMER software was used. The second stage ranked the best energy alternative for a mid-rise building in Egypt using AHP and Technique for Order Preference by Similarity to an Ideal Solution (TOPSIS). Kotb et al. [37] used a decision-making model to determine the best energy option for an Egyptian resort. The techno-economic characteristics of the various alternatives were developed using HOMER software. To rank and choose the best configuration, a combination of Fuzzy-AHP and Fuzzy-VIKOR multi-attributes decision-making techniques were used.

Based on the literature reviewed above, it can be concluded that the majority of studies focused on the economic features to determine and rank the best system. For those who used MADM methods [38–40], no explanation was given as to why a particular approach to weight assignment was used.

The contribution of this study can be stated as follows:

- Developing a dual-phase optimization approach among various power system options, including standalone generators, hybrid PV-Diesel-Wind-Battery, and hybrid

PV-Biogas-Wind-Battery, in which HOMER is used in the first phase to optimally design the various configurations, and an MADM technique is used in the second phase to rank the optimal configuration.
- Using two separate approaches to assign weights to the selected attributes in order to see how they affect the decision-making process.
- Comparing the efficiency of various backup components in order to achieve a secure power supply strategy.
- Doing a sensitivity analysis on the optimum configuration to assess the effects of changing input parameters.

## 2. Research Methodology

### 2.1. Area Description, Load Demand, and Available Resources

The area under consideration is Banana Islands, a small coastal community in Sierra Leone's western region, situated southwest of the Freetown peninsula. The island is geographically located at 8°7.0′ N, 13°12.7′ W as seen in Figure 1. Rickets, Dublin, and Mes-Meheux are the three main islands that make up the archipelago. Ricket is known for its forest, Dublin is known for its beaches, and Mes-Meheux is known for its adventurous tourism [41]. The climate in the Banana Islands varies somewhat, but it is generally hot throughout the year with little chance of rain. The temperature ranges from 25.6 °C (78 °F) to 31 °C (87.8 °F) according to [42]. The island is powered by a 155 kW generator that runs between the hours of 6:00 p.m. and 10:00 p.m. Since travelers must use paying boats and ferries to move from and to the island, diesel is slightly more expensive than on the mainland. In comparison to grid-connected regions, this raises the cost of electricity significantly. For the four hours of electricity given, each household pays around $0.6 a day. The island has a total population of 900 people living in 200 households. Per household's average daily electric consumption is 2.45 kW-hours, resulting in a total daily consumption of 490 kWh/day, with a peak load of 68.18 kW as seen in Figure 2. Peak load is usually observed in the evenings around 6:00 p.m., when the majority of the residents have returned from work. Owing to the Christmas and New Year's holidays, visitors begin to flock to the area from November to January. During this time, the load demand rises. The resources used in this analysis are solar, wind, and biomass. The monthly average temperature, solar radiation, and clearness index and wind speed were downloaded from NASA's website as shown in Figure 3A,B. The annual average solar radiation, wind speed, and ambient temperature of the case study over a period of 22 years are 5.34 kWh/m$^2$/day, 3.42 m/s, and 26.43 °C. The solar radiation and clearness index increases during the dry season between January and April, with March being the peak month, as shown in Figure 3A. It decreases during the rainy season, with the lowest solar production in July and August. Wind, on the other hand, has been observed to increase from June to September. Agricultural activities are carried out on the island especially in Ricket that is surrounded by dense forest. Agriculture is practiced on the island, especially in Ricket, which is surrounded by dense forest. Rice farming is the region's primary source of income. The biomass material for the biogas generator is rice husk. According to [6], a similar study on rural electrification in Sierra Leone, the average amount of rice husk available per day is 3.27 tonnes. Carbon content of the rice husk is 42.60%, gasification ratio is 39.21%, and low heat value (LHV) is 14.12 MJ/kg.

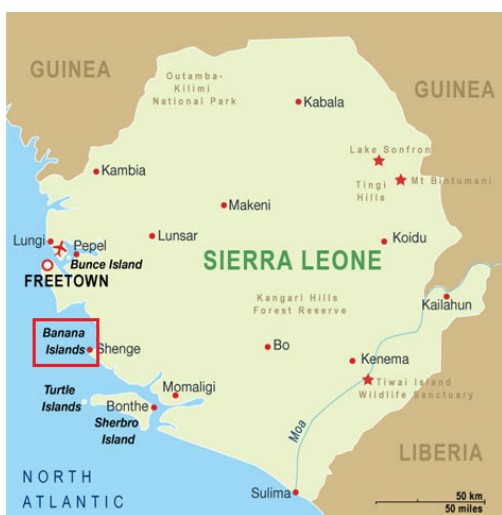

**Figure 1.** Map of Banana Islands [43].

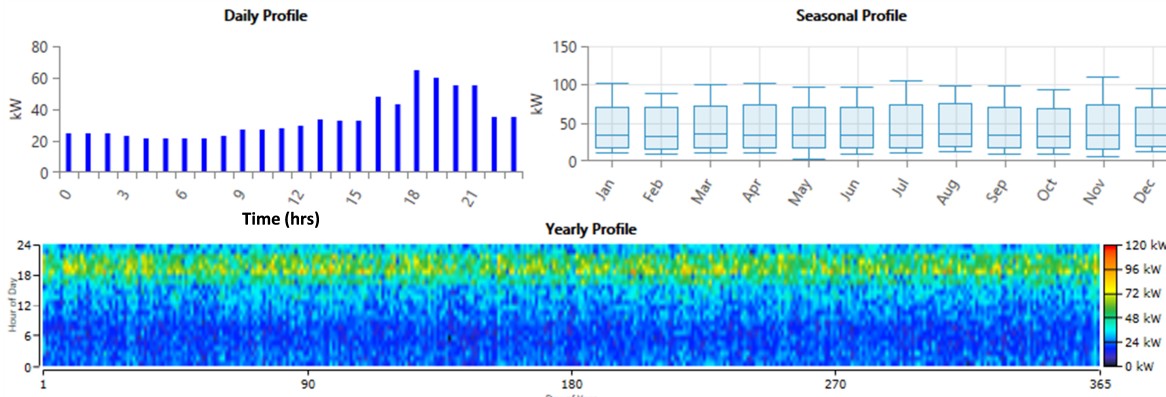

**Figure 2.** Daily and seasonal load profile.

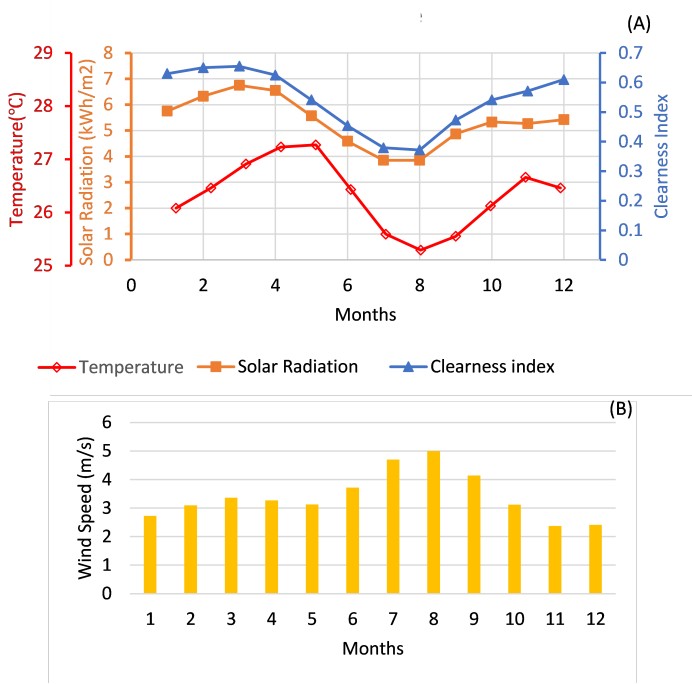

**Figure 3.** Average monthly: (**A**) solar radiation, temperature, and clearness index; (**B**) wind speed.

### 2.2. System Component and Mathematical Modelling

This research looks at six different components: a diesel generator, a biogas generator, a wind turbine, a photovoltaic module, a battery, and a converter. Figure 4 shows a detailed component schematic. Table 1 establishes both the technical and economic parameters of the various components. Table 2 shows the search space and HOMER optimizer parameters for various components.

#### 2.2.1. Modeling of Photovoltaic Module

The Sharp PV module was chosen for this study because it is readily available in Sierra Leone and neighboring countries. During the day, from 7:00 a.m. to 6:00 p.m., the PV generates energy directly from the sun. The power output of the PV module is measured using Equation (1), which is referenced to [44], taking into account temperature effects on the PV module:

$$p_{p_v} = W_{p_v} f_{p_v} G_{T/G_s} \left[ 1 + K_p (T_c - T_{STC}) \right] \tag{1}$$

where $W_{p_v}$ is the peak power output (kW), $f_{p_v}$ is the derating factor, $G_T$ is the solar radiation on PV at a specific time (W/m$^2$), $G_s$ is the standard test irradiation (1000 W/m$^2$), $K_p$ is the temperature coefficient of power (%/°C), $T_c$ is the PV module temperature at hour (°C), and $T_{STC}$ is the PV module temperature under standard conditions (298 K).

For a PV to produce the desired output, the ambient temperature is critical. This ambient temperature must not exceed the appropriate temperature, which can only be accomplished by correctly aligning the PV panel with the sun's rays. Both inadequate and increased solar radiation have a significant impact on the PV panel's derating factor. In order to stay within a safe range, the equation for calculating the surface temperature of the PV panel [45] is given in Equation (2):

$$T_s = T_a + T_{s,n} - T_{a,n} \left( \frac{G_T}{G_{T,n}} \right) \left( 1 - \frac{\eta_{mp}}{\beta \tau} \right) \tag{2}$$

where $T_a$ is the ambient temperature (°C), $T_{s,n}$ is the cell's nominal operating temperature (°C), $T_{a,n}$ is ambient temperature in the NOCT condition (20°), $G_{T,n}$ is the sun's radiation in NOCT condition, $\eta_{mp}$ is the efficiency of the panel, $\tau$ is the solar transmittance, and $\beta$ is the solar panel absorption.

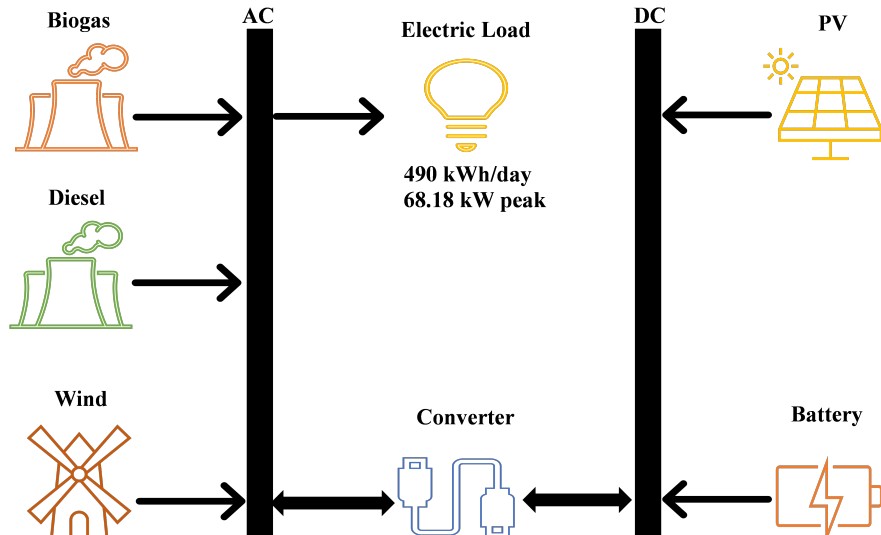

**Figure 4.** Component schematic.

**Table 1.** Techno-economic characteristics of the various components.

| Reference | Description | Specification |
|:---:|:---:|:---:|
| [8] | **PV system** | |
| | PV type | Sharp ND-250 QCS |
| | Capital cost ($) | 1300 |
| | Replacement cost ($) | 1300 |
| | O&M cost ($/yr) | 10 |
| | Lifetime (years) | 25 |
| [8] | **Converter** | |
| | Capital cost ($) | 600 |
| | Replacement cost ($) | 600 |
| | O&M cost ($/yr) | 10 |
| | Lifetime (years) | 15 |
| [2] | **Diesel Generator** | |
| | Capital cost ($) | 660 |
| | Replacement cost ($) | 660 |
| | O&M cost ($/yr) | 0.03 |
| | Lifetime (hours) | 15,000 |
| [8] | **Biogas Generator** | |
| | Capital cost ($) | 1500 |
| | Replacement cost ($) | 1500 |
| | O&M cost ($/yr) | 0.01 |
| | Lifetime (hours) | 20,000 |
| [37] | **Wind turbine** | |
| | Turbine type | Eocycle E025 Class 111 |
| | Capital cost ($) | 40,000 |
| | Replacement cost ($) | 36,000 |
| | O&M cost ($/yr) | 500 |
| | Lifetime (years) | 20 |
| | Hub height (m) | 23 |
| [2] | **Battery** | |
| | Battery type | Hoppecke 24 OPzS 3000 |
| | Capital cost ($) | 1259 |
| | Replacement cost ($) | 1000 |
| | O&M cost ($/yr) | 10 |
| | Lifetime (years) | 20 |
| | Nominal capacity (kWh) | 7.15 |

**Table 2.** Search space and HOMER optimizer parameters.

| PV Array (kW) | Wind Turbine (No. ) | Biogas Genset (kW) | Diesel Genset (kW) | Battery (kWh) | Converter (kW) |
|:---:|:---:|:---:|:---:|:---:|:---:|
| 0–400 | 1–5 | 0 | 0 | 0–300 | 0–150 |
| | | 30 | 30 | | |
| | | 50 | 50 | | |
| | | 75 | 75 | | |
| | | 100 | 100 | | |

### 2.2.2. Modeling of the Battery Storage System

Batteries aid in the promotion of power supply stability, efficiency, and reliability, especially when renewable energy sources are unavailable. They can also be used to cut down on fuel consumption. As seen in Equations (3) and (4), storage importance is linked to the effectiveness of its charging and discharging abilities [45]

Charging Process:

$$E_{b_i}(t) = E_{b_i}(t-1)(1-\sigma) + \{(E_{b_ih}(t) - E_{b_il}(t)/\eta_{b_j} \times \eta_{b_b}\} \tag{3}$$

Discharging Process:

$$E_{b_i}(t) = E_{b_i}(t-1)(1-\sigma) - (E_{b_ih}(t)/\eta_{b_j} - E_{b_il}(t)), \tag{4}$$

where $E_{b_i}$ is the net battery energy in time (t), $E_{b_ih}$ is the total energy generated at a specific time, $E_{b_il}$ is the load demand in time (t), $\eta_{b_j}$ is the converter efficiency, $\eta_{b_b}$ is the charging efficiency of the battery, and $\sigma$ is the rate of self discharge. For a battery to retain enough charge to support a designed system when PV powers are unavailable, the capacity must be tailored by using Equation (5) which is referenced to [16].

$$C_{BAT} = \left[\frac{E_{load(t).DA(t)}}{\eta_{con}.\eta_{BAT}.DOD(t)}\right] \tag{5}$$

where $E_{load}$ is the average energy demand (kWh/day), DA is the days of autonomy, $\eta_{con}$ is the efficiency of converter, $\eta_{BAT}$ is the efficiency of battery, and DOD is the depth of charge of the battery.

### 2.2.3. Modeling of Biogas Generator

However, there are two distinct ways to generate electricity from biomass: either by thermochemical means such as gasification or combustion, or by a biochemical mechanism such as fermentation [46]. This research considers the gasification process. Due to its availability and the fact that agriculture is the primary occupation of the residents of the case study settlement, rice husk is used as a biomass resource. Equation (6), which is cited in [29], can be used to measure the output power from a biogas generator:

$$P_{BG}(t) = \left(\frac{N_{gas}}{F_1}\right)\left[\frac{\eta_{gas} \times H_a \times bio(t)}{H_{gas}} - F_0 P_e\right] \tag{6}$$

where $F_0$ is intercept coefficient = 0.1 (kg/hr/kW) and $F_1$ is the slope = 2 (kg/hr/kW) for a 100-kW generator. The size of the biogas generator used in the simulation process determines the values of $F_0$ and $F_1$. The component's range is 0–100 kW, according to Table 2. As a result, the maximum component size is 100 kW, and the values of $F_0$ and $F_1$ are based on the biogas generator's maximum output capacity.

### 2.2.4. Modeling of Wind Turbine

The speed of a wind turbine at the hub height in the chosen region has a significant impact on its power production. As a result, Equation (7) is used to measure speed at hub height. Equation (8), which is referenced to [47], can then be used to measure the output power of wind turbines:

$$X_{hub} = X_{anem}\frac{ln(\frac{z_{hub}}{z_0})}{ln(\frac{z_{anem}}{z_0})} \tag{7}$$

$$P_{WTG} = P_{WTG,STP}\left(\frac{\rho}{\rho_{STP}}\right) \tag{8}$$

where $X_{hub}$ is wind speed at hub height, $z_{hub}$ is the hub height of wind turbine, $z_{anem}$ is the anemometer height, $z_0$ is surface roughness length, $P_{WTG}$ is the power output of wind turbine, $P_{WTG,STP}$ is the turbine power output at standard temperature and pressure, $\rho$ is the air density, and $\rho_{STP}$ is the air density at STP.

### 2.2.5. Hybrid Configurations

In this study, six hybrid configurations were modeled. The study looked at two types of backup generators: diesel and biogas generators. The hybrid PV-diesel-wind-battery configuration yielded three hybrid configurations, while the PV-biogas-wind-battery configuration also yielded three hybrid configurations. The aim of the separate backup components is to see which of the two (diesel or biogas) is the better choice for complementing the PV and wind resources' intermittent existence. PV-Wind-Diesel-Battery (PV+W+D+B), PV-Diesel-Battery (PV+D+B), Wind-Diesel-Battery (W+D+B), PV-Wind-Biogas-Battery (PV+W+BG+B), PV-Biogas-Battery (PV+BG+B), and Wind-Biogas-Battery (W+BG+B) are the different configurations.

### 2.3. System Optimization

The flowchart of the research methodology is shown in Figure 5. HOMER modeling tool is used in the first phase of this study. It has been used to model hybrid energy sources all over the world because it is effective in both technical and economic terms. Its applications have been widely used in both off-grid and grid-connected systems to properly plan, conduct feasibility studies, schedule, and evaluate system inputs in order to provide the best solution for a sustainable energy climate [2]. The research's goal is to reduce the NPC, which is provided in Equation (9) and referenced in [47]:

$$Minimize(NPC) = \frac{C_{ann,tot}}{CRF(i, R_p)} \tag{9}$$

$$CRF(i, R_p) = \frac{i(1+i)^n}{(1+i)^n - 1} \tag{10}$$

where $C_{ann,tot}$ is the total annualized cost (\$/yr), *CRF* is the capital recovery factor as calculated using Equation (10), *i* is the annual interest rate. An interest rate of 8% was used in the simulation process. $R_p$ is the project lifetime. The sizing, techno-economic, and environmental results from HOMER software are used as the main inputs in the MADM technique (CODAS algorithm).

In the second step, a MADM technique is used. They've been widely used in the optimization of hybrid renewable energy systems, where there are several attributes to consider in achieving a specific goal or set of goals, and hence they seem to clash. They've been shown to be effective tools for deciding between a variety of competing options [48]. CODAS is the key decision-making technique used in this study to rate the various configurations. It's a scoring-based MADM approach that's used in the second phase of the optimization process to rank the best configuration for a long-term, cost-effective power system. In order to determine which strategy is best for choosing the optimum configuration, this study considers two separate methods of assigning weights to the attributes. The subjective method (AHP), introduced by Saaty [49], is the first method. The weights of the attributes are determined using this approach, which is based on the decision maker's experience and preferences. The objective method (Entropy method) is the second method, which uses mathematical applications to evaluate the weights of the attributes based on the objective decision matrix details. Here, the decision maker has no role in determining the importance of the attributes [33]. The attributes considered are divided into economic: NPC, COE, and operation and maintenance cost (O&M); technical: Excess electricity and Renewable fraction and environmental: Carbon dioxide ($CO_2$ emissions). The various steps used in the execution of the CODAS procedure are given in Algorithm 1. Algorithm 2 explains the procedures for calculating the various weights using the AHP approach. Algorithm 3 explains the Entropy weight calculation procedures.

---

**Algorithm 1:** CODAS algorithm steps.

---

1:    Select the required attributes

2:    Build the decision matrix

3:    Normalization of the decision matrix

$$\begin{cases} \dfrac{K_{ij}}{max_i K_{ij}} & if \ j\epsilon N_b \\[2mm] \dfrac{min_i K_i}{K_{ij}} & if \ j\epsilon N_{nb} \end{cases} \tag{11}$$

where $N_b$ is for beneficial attributes and $N_{nb}$ is for non-beneficial attributes.

4:    Determine the Euclidean ($E_i$) and Taxicab ($T_i$) distances

$$E_i = \sqrt{\sum_{m}^{j=1} \left(r_{ij} - ns_j\right)^2}, \ \ T_i = \sum_{m}^{j=1} \left|r_{ij} - ns_j\right| \tag{12}$$

5:    Estimate the weighted normalized matrix

$$r_{ij} = w_j n_{ij}, \ 0 < w_j < 1, \ \sum_{n}^{j} w_j = 1 \tag{13}$$

6:    Calculate the negative idea solution

$$ns = \left[ns_j\right]_{1 \times m}, \ ns_j = min_i r_{ij} \tag{14}$$

7:    Determine the Euclidean ($E_i$) and Taxicab ($T_i$) distances

$$E_i = \sqrt{\sum_{m}^{j=1} \left(r_{ij} - ns_j\right)^2}, \ \ T_i = \sum_{m}^{j=1} \left|r_{ij} - ns_j\right| \tag{15}$$

8:    Construct the relative assessment matrix

$$R_a = \left[(E_i - E_k) + (\psi(E_i - E_k) \times (T_i - T_k)), \ k\epsilon\{1, 2...., n\}\right] \tag{16}$$

$$\psi(x) = \begin{cases} 1 \ if \ |x| \geq \tau \\ 0 \ if \ |x| < \tau \end{cases} \tag{17}$$

where $\tau$ stands for the threshold value that is independent of every decision maker. It lies in the range of
0.01 to 0.05. In this research, we used 0.02.

9:    Calculate the assessment score

$$H_i = \sum_{n}^{k=1} h_{ik} \tag{18}$$

10:   The various alternatives are ranked from highest to lowest value of the assessment score. The best alternative is the one with the highest score.

---

| **Algorithm 2:** AHP procedures and mathematical relations for obtaining the weights. |
|---|
| 1:　Build the hierarchy structure of the decision-making process |
| 2:　Create using an expert's opinion a pairwise comparison matrix C, where |
| 　　C = $c_{ij}$ is defined as element of |
| 　　decision matrix in i-th column and j-th row |
| 3:　Produce the average weighting factor for each attribute/criterion by taking the average in j-th row |
| 4:　Calculate the consistency index (CI) and consistency ratio (CR) where |
| 　　CI = $(\lambda_{max} - n)/(n-1)$, CR = CI/RI; |
| 　　of which RI is a random index that depends |
| 　　on "n," where "n" is the number of alternatives/configurations |

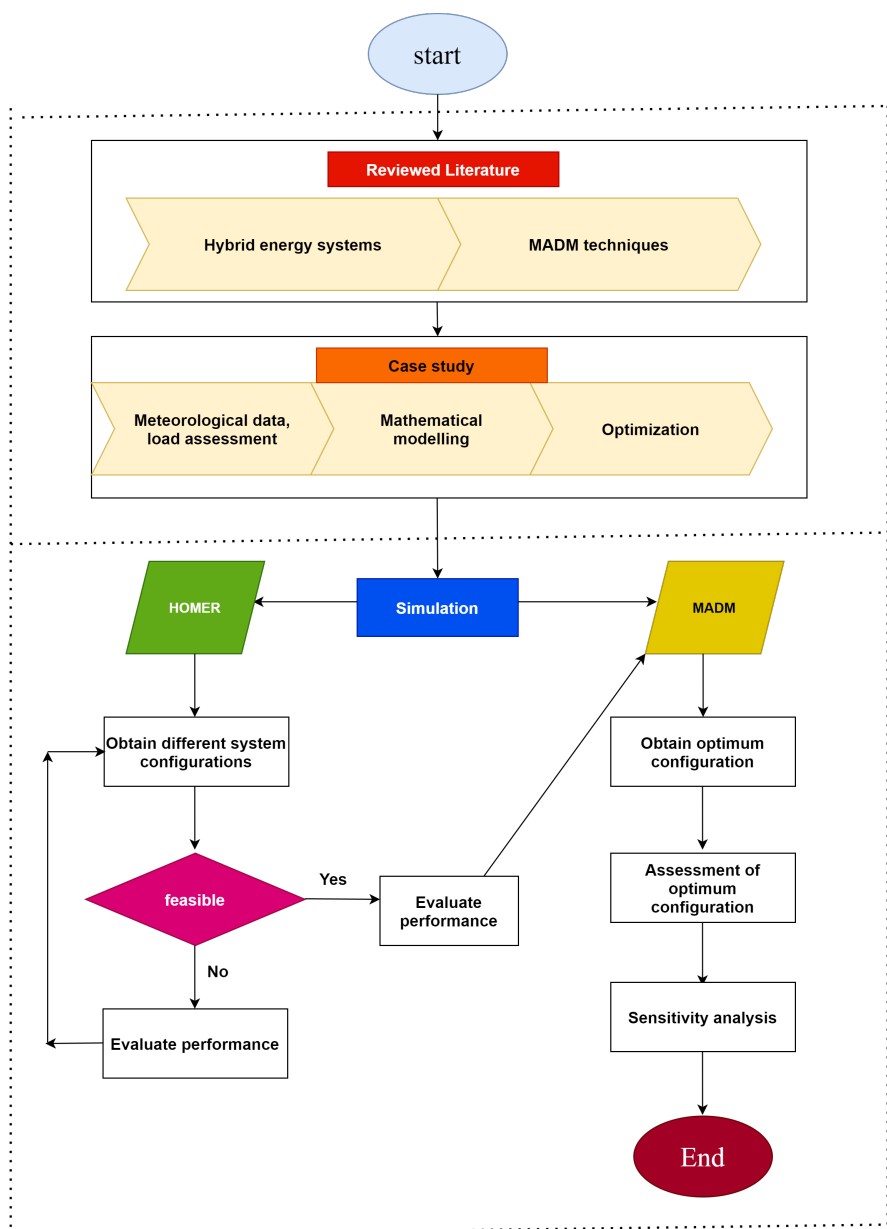

**Figure 5.** Flowchart of research methodology.

---

**Algorithm 3:** Entropy procedures and mathematical relations for obtaining the weights.

1:   Normalization of the decision matrix $p_{ij}$

$$P_{ij} = \frac{X_{ij}}{\sum_{i=1}^{m} X_{ij}} \tag{19}$$

2:   Compute the measure of the entropy

$$E_j = -K \sum_{i=1}^{m} P_{ij} ln P_{ij}, \ K = 1/ln(m)) \tag{20}$$

3:   Define the objective weights based on the concept of entropy

$$w_j = \frac{1 - E_j}{\sum_{j=1}^{n} (1 - E_j)} \tag{21}$$

---

## 3. Results and Discussion

### 3.1. Result of the First Optimization Approach

The results of the first phase of the study are shown in Table 3. It shows the component sizing, technical, economic, and environmental characteristics of the current base case as well as the six hybrid renewable energy configurations that used both diesel and biogas generators as backup components to supplement the solar and wind resources' inconsistencies. The optimization results show that a standalone diesel configuration is not a good choice for providing electricity to the island 24 h a day, 7 days a week. Despite the fact that it can handle the load, it is the least cost-effective and environmentally friendly configuration. It has the highest COE ($0.598/kWh), NPC ($1,382,532), O&M ($254,801.35), and $CO_2$ emission (152,707 kg/yr). This design is not feasible for the island because the primary goal of this study is to reduce overall system costs.

Due to their low unmet loads, which range from 0.01% to 0.07%, all six hybrid configurations are considered reliable options from a technical standpoint. The W+DG+B configuration provides the least amount of excess electricity (9172 kWh/yr) and the second least amount of unmet load (52.4 kWh/yr), but it has the lowest renewable fraction (27.3%). The low excess electricity is due to the low renewable penetration, which stems from the wind turbine's low contribution. Due to the increased renewable fraction (88.3%), the PV+DG+B configuration generates the least unmet load (24.7 kWh/yr) but also produces the most excess electricity (43,393 kWh/yr). The PV+W+BG+B configuration is the only one with a 100% renewable penetration while still producing a fair amount of excess electricity (27,763 kWh/yr), but it falls short due to the high unmet load (132 kWh/yr). When compared to diesel-backup configurations, biogas generator backup configurations provided more unmet loads and excess electricity. This is due to a rise in the use of renewable energy. We can now conclude that the PV+DG+B configuration is the most efficient, with the lowest unmet load, despite having a higher excess electricity and a modest renewable fraction.

The PV+W+BG+B configuration is the most cost-effective due to its low financial records. The NPC is $487,247, and COE is $0.211/kWh. It also has the cheapest O&M of $41,502.13. The low financial records are the result of smaller component sizes (1 wind turbine, 101 kW of PV, 50 kW biogas generator, 86 batteries, and 37.6 kW converter ). The W+DG+B configuration has the lowest initial capital cost of the hybrid configurations. Among the different hybrid configurations, the W+BG+B configuration has the worst economic records. This is because there are a lot of batteries (187 batteries) and wind turbines (16 turbines) used. One of the drawbacks of using a larger number of wind turbines and a biogas generator with longer operating hours is the high cost of operation and mainte-

nance. The PV+W+BG+B configuration reduces the NPC by 64.7% and 59.6%, respectively, as compared to the current standalone diesel configuration and the W+BG+B configuration.

**Table 3.** Optimization results of the various hybrid configurations.

| Configuration | Units | Disel-Based | Renewable-Diesel Backup | | | Renewable-Biogas Backup | | |
|---|---|---|---|---|---|---|---|---|
| | | | PV+W+DG+B | PV+DG+B | W+DG+B | PV+W+BG+B | PV+BG+B | W+BG+B |
| **Optimal sizing** | | | | | | | | |
| WTs | No. | - | 2 | - | 2 | 1 | - | 16 |
| PV | kW | - | 109 | 143 | - | 101 | 141 | - |
| DG | kW | 75 | 30 | 50 | 30 | - | - | - |
| BG | kW | | - | - | - | 50 | 50 | 75 |
| BAT | No. | | 103 | 84 | 31 | 86 | 86 | 187 |
| CON | kW | | 48.1000 | 49.3000 | 31.8000 | 37.6000 | 37.6000 | 66.9000 |
| **Technical** | | | | | | | | |
| Excess elect. | kWh/yr | 22,124 | 48,048 | 43,393 | 9172 | 27,763 | 50,998 | 378,363 |
| Ren. fract. | % | 0 | 96.8000 | 88.3000 | 27.3000 | 100 | 100 | 100 |
| Unmet load | kWh/yr | 0 | 101 | 24.70000 | 52.40000 | 132 | 121 | 143 |
| **Economic** | | | | | | | | |
| NPC | $ | 1,382,532 | 545,436 | 588,211 | 702,897 | 487,247 | 530,857 | 1,280,000 |
| COE | $/kWh | 0.598 | 0.2360 | 0.2540 | 0.3040 | 0.2110 | 0.2300 | 0.5520 |
| Initial capital | $ | 49,500 | 400,359 | 354,749 | 157,884 | 376,701 | 408,781 | 1,003,000 |
| O&M cost | $ | 254,801.3500 | 52,412.2900 | 60,844.7800 | 72,117.1700 | 41,502.1300 | 42,633.6700 | 140,367.3100 |
| **Environment** | | | | | | | | |
| $CO_2$ | kg/yr | 152,707 | 4200 | 15,763 | 88,059 | 17.5000 | 16.6000 | 9.2200 |

When compared to their diesel-backup counterparts, all configurations that used the biogas generator as a backup component emitted low $CO_2$ emissions. Despite the fact that the W+BG+B configuration tends to be the worst in terms of techno-economic features, it is the most environmentally friendly, emitting 9.22 kg $CO_2$ per year, followed by PV+BG+B (16.6 kg/yr) and PV+W+BG+B (17.5 kg/yr). The PV+W+BG+B configuration emits more $CO_2$ than the W+BG+B configuration due to the increased biogas activity to complement the number of batteries. Among the hybrid configurations, the W+DG+B generates the most $CO_2$ emissions (88,059 kg/yr). The reason for this is that wind resources have strong intermittencies. To compensate for the lack of wind, the diesel generator runs for longer periods of time. This increased activity results in more poisonous gas emissions.

According to the above analyses, no single hybrid configuration outperforms the others in terms of technical, economic, and environmental characteristics. The PV+DG+B configuration provided the lowest unmet load, making it technically sound, but it falls short in terms of economic and environmental features when compared to the PV+W+BG+B and W+BG+B configurations. This makes choosing the best configuration extremely difficult. It would be a biased decision to choose any of the configurations as the best, based on the results of this first optimization. Since we are considering multiple attributes with competing interests, multi-attribute decision-making could be a better strategy.

*3.2. Result of the Second Optimization*

Two separate weight assignment methods were considered in this phase. The first has to do with the AHP weight assignment and the second considered Entropy weight assignment. The steps involved in the various weights calculation are explained in Section 2.3.

3.2.1. AHP-CODAS Approach

The first optimization's results were used as input data for the MADM operation. Table 4 shows the different weights obtained using the AHP algorithm. It shows that, for a developing country like Sierra Leone, COE and NPC had the highest priorities with scores of 20.80% and 34.16%, respectively. The negative sign next to an attribute means that it should be diminished, while the positive sign indicates that it should be maximized. The higher the renewable fraction, the cleaner the system is for the atmosphere, while

lowering the cost parameters increases the island's socio-economic status. Considering steps 1 and 2 of implementing the CODAS algorithm, the system attributes and initial decision matrix are presented in Table 5. Results for the Normalized matrix of the CODAS procedure are given in Table 6. Both the AHP and Entropy weight assignment strategies would use it as an input to calculate the weighted normalized matrix. The AHP-CODAS approach's weighted normalized matrix is obtained by multiplying the different weights with the normalized matrix. Table 7 displays this information. Calculations from negative ideal solution and Euclidean and taxicab distances (steps 5 and 6) resulted in the formation of the relative assessment matrix. The relative assessment matrix, assessment score, and rank of the various configurations of the AHP-CODAS approach are presented in Table 8. Because of the highest assessment score, the PV-W-BG-B configuration is ranked as the best. Table 3 shows that this design has a 100% renewable fraction as well as the lowest NPC ($487,247) and COE ($0.211/kWh). It has 1 wind turbine, 101 kW PV, 50 kW biogas generator, 86 batteries, and a 37.6 kW converter in its system configuration. Despite having the highest unmet load and NPC, the W-BG-B is ranked second. With an annual $CO_2$ emission of 9.22 kg/yr, it is considered the most environmentally friendly configuration. The first three rated configurations used biogas as a backup, recognizing the superiority of using a biogas generator as a backup component compared to those that used diesel as backup.

**Table 4.** Weights of AHP analysis.

| Attribute | Type | Weight | % |
|---|---|---|---|
| Cost of energy (COE) | Non-beneficial (−) | 0.2080 | 20.80 |
| Net present cost (NPC) | Non-beneficial (−) | 0.3416 | 34.16 |
| Renewable fraction (RF) | Beneficial (+) | 0.1401 | 14.01 |
| Excess electricity (EXE) | Non-beneficial (−) | 0.1765 | 17.65 |
| Carbon emission gases (CO2) | Non-beneficial (−) | 0.1338 | 13.38 |

**Table 5.** Attributes and initial decision matrix.

| Configurations | NPC | COE | RF | EXE | $CO_2$ |
|---|---|---|---|---|---|
| PV-W-DG-B | 0.2360 | 545,436 | 96.8000 | 48,048 | 4200 |
| PV-DG-B | 0.2540 | 588,211 | 88.3000 | 43,393 | 15,763 |
| W-DG-B | 0.3040 | 702,897 | 27.3000 | 9172 | 88,059 |
| PV-W-BG-B | 0.2110 | 487,247 | 100 | 27,763 | 17.5 |
| PV-BG-B | 0.2300 | 530,857 | 100 | 50,998 | 16.6 |
| W-BG-B | 0.5520 | 1,280,000 | 100 | 378,363 | 9.22 |

**Table 6.** Normalized matrix of the CODAS Algorithm.

| Configuration | COE | NPC | RF | EXE | $CO_2$ |
|---|---|---|---|---|---|
| PV-W-DG-B | 0.8941 | 0.8933 | 0.9680 | 0.1909 | 0.0022 |
| PV-DG-B | 0.8307 | 0.8284 | 0.8830 | 0.2114 | 0.0006 |
| W-DG-B | 0.6941 | 0.6932 | 0.2730 | 1.0000 | 0.0001 |
| PV-W-BG-B | 1.0000 | 1.0000 | 1.0000 | 0.3304 | 0.5269 |
| PV-BG-B | 0.9174 | 0.9178 | 1.0000 | 0.1799 | 0.5554 |
| W-BG-B | 0.3822 | 0.3807 | 1.0000 | 0.0242 | 1.0000 |

**Table 7.** Weighted normalized matrix for the AHP-CODAS approach.

| Configuration | COE | NPC | RF | EXE | $CO_2$ |
|---|---|---|---|---|---|
| PV-W-DG-B | 0.3054 | 0.1858 | 0.1356 | 0.0337 | 0.0003 |
| PV-DG-B | 0.2838 | 0.1723 | 0.1237 | 0.0373 | 0.0001 |
| W-DG-B | 0.2371 | 0.1442 | 0.0382 | 0.1765 | 0.0000 |
| PV-W-BG-B | 0.3416 | 0.2080 | 0.1401 | 0.0583 | 0.0705 |
| PV-BG-B | 0.3134 | 0.1909 | 0.1401 | 0.0317 | 0.0743 |
| W-BG-B | 0.1306 | 0.0792 | 0.1401 | 0.0043 | 0.1338 |

**Table 8.** Assessment matrix, score, and ranking of the configurations (AHP-CODAS approach).

| Configurations | COE | NPC | RF | EXE | $CO_2$ | Assessment Score | Rank |
|---|---|---|---|---|---|---|---|
| PV-W-DG-B | 0.0000 | 0.0120 | 0.0073 | −0.0920 | −0.0786 | −0.1514 | 4 |
| PV-DG-B | −0.0120 | 0.0000 | −0.0046 | −0.1039 | −0.0905 | −0.2110 | 6 |
| W-DG-B | −0.0073 | 0.0046 | 0.0000 | −0.0992 | −0.0858 | −0.1877 | 5 |
| PV-W-BG-B | 0.0926 | 0.1047 | 0.1001 | 0.0000 | 0.0136 | 0.3110 | **1** |
| PV-BG-B | 0.0789 | 0.0910 | 0.0864 | −0.0135 | 0.0000 | 0.2427 | 3 |
| W-BG-B | 0.0913 | 0.1033 | 0.0987 | −0.0007 | 0.0128 | 0.3053 | 2 |

### 3.2.2. ENTROPY-CODAS Approach

The attributes and initial decision matrix are presented in Table 5. Results of the Normalized matrix of the CODAS procedure are given in Table 6. The weights obtained from the Entropy calculations are given in Table 9. The highest weights were assigned to the $CO_2$ emissions and excess electricity. The weighted normalized matrix of the Entropy-CODAS approach is presented in Table 10. The relative assessment matrix, assessment score, and rank of the various configurations of the Entropy-CODAS approach are presented in Table 11. The W-BG-B system is ranked as the best configuration because of the highest assessment score. The PV-BG-B is ranked second and the PV-W-BG-B follows in third place. The success of the W-BG-B configuration is due to the initial preference weight given to the $CO_2$ emission. This result also confirms the superiority of the configurations that used the biogas generator as a backup component by order of ranking compared to those that used diesel as a backup.

**Table 9.** Weights of entropy analysis.

| Attribute | Type | Weight | % |
|---|---|---|---|
| Cost of energy (COE) | Non-beneficial (−) | 0.0319 | 3.19 |
| Net present cost (NPC) | Non-beneficial (−) | 0.0322 | 3.22 |
| Renewable fraction (RF) | Beneficial (+) | 0.0292 | 2.92 |
| Excess electricity (EXE) | Non-beneficial (−) | 0.3264 | 32.64 |
| Carbon emission gases ($CO_2$) | Non-beneficial (−) | 0.5803 | 58.03 |

**Table 10.** Weighted normalized matrix for the Entropy-CODAS approach.

| Configuration | COE | NPC | RF | EXE | $CO_2$ |
|---|---|---|---|---|---|
| PV-W-DG-B | 0.0285 | 0.0285 | 0.0282 | 0.0623 | 0.0013 |
| PV-DG-B | 0.0265 | 0.0266 | 0.0257 | 0.0690 | 0.0003 |
| W-DG-B | 0.0221 | 0.0223 | 0.0079 | 0.3264 | 0.0001 |
| PV-W-BG-B | 0.0319 | 0.0321 | 0.0291 | 0.1078 | 0.3057 |
| PV-BG-B | 0.0293 | 0.0295 | 0.0291 | 0.0587 | 0.3223 |
| W-BG-B | 0.0122 | 0.0122 | 0.0291 | 0.0079 | 0.5803 |

**Table 11.** Assessment matrix, score, and ranking of the configurations (Entropy-CODAS approach).

| Configurations | COE | NPC | RF | EXE | CO$_2$ | Assessment Score | Rank |
|---|---|---|---|---|---|---|---|
| PV-W-DG-B | 0.0000 | 0.0004 | −0.0961 | −0.3095 | −0.3187 | −0.7239 | 5 |
| PV-DG-B | −0.0004 | 0.0000 | −0.0965 | −0.3099 | −0.3191 | −0.7258 | 6 |
| W-DG-B | 0.0970 | 0.0973 | 0.0000 | −0.2147 | −0.2238 | −0.2442 | 4 |
| PV-W-BG-B | 0.3140 | 0.3144 | 0.2158 | 0.0000 | 0.0090 | 0.8352 | 3 |
| PV-BG-B | 0.3228 | 0.3232 | 0.2246 | −0.0090 | 0.0000 | 0.8796 | 2 |
| W-BG-B | 0.5812 | 0.5816 | 0.4816 | −0.2645 | 0.2557 | 2.1647 | **1** |

### 3.2.3. Comparative Analysis of Weight Assignment Approaches

The AHP method of assigning weights is robust and superior to the Entropy method, as shown by the results of the ranking of the different configurations in Tables 8 and 11. The main goal of this study is to find a cost-effective and long-term power system design for supplying electricity to Sierra Leone's Banana Islands. The PV-W-BG-B configuration is the most cost-effective, with the lowest NPC, COE, and O&M costs, while the W-BG-B configuration is the least cost-effective. The PV-W-BG-B configuration has lower NPC, COE, and O&M costs than the W-BG-B configuration by 61.9%, 61.7%, and 70.4%, respectively. When opposed to the W-BG-B configuration, the PV-W-BG-B configuration generates 7.69% less unmet load and 92.6% less excess electricity, making it more capable and effective. When compared to the PV-W-BG-B configuration, the W-BG-B configuration emits 47.3% less CO$_2$ emissions. Since our objective is to get a configuration that is cost-effective and sustainable, the PV-W-BG-B configuration is preferable. This proves that the AHP-weight assignment method is superior to the Entropy-weight assignment method. The AHP method of weight assignment is based on expert judgment, and is a trustworthy method since it allows decisions based on the decision maker's previous knowledge and growth needs. The decision outcome is focused on the objective decision matrix knowledge using mathematical applications, and there is no space for expertise in the Entropy analysis.

### 3.2.4. Performance Assessment of the Optimum Configuration

Figures 6 and 7 show the technical analyses in greater detail. The monthly electric outputs of the PV, wind, and biogas components are shown in Figure 6. The PV panel dominates electricity generation (70.3%), led by the wind turbine (15.6%) and the biogas generator (14.1%). PV power generation increases in the dry season (November to April) due to a high clearness index and decreases in the rainy season (June to September) due to dark clouds and heavy rains, as observed. In July and August, when the wind speed in the area rises, the wind contributes the most electricity. The biogas contribution is highest in November and December which is when the island experiences the lowest wind speed. To compensate for the wind's inconsistencies, the biogas generator increases its contribution. The battery efficiency is shown in Figure 7. The expected lifespan is 11.7 years, with a 21.1-h autonomy. Discharge is most noticeable in the mornings, when solar energy is scarce. The cost description of the different components is shown in Figure 8. The battery is the most expensive component of the system, followed by PV, biogas, wind, and the converter. As shown in Figure 9, the high battery cost is due to the fact that it is replaced twice before the project life expires. Environmentally, the optimum system produces a very low amount CO$_2$ compared to the existing stand-alone diesel configuration as seen in Table 12.

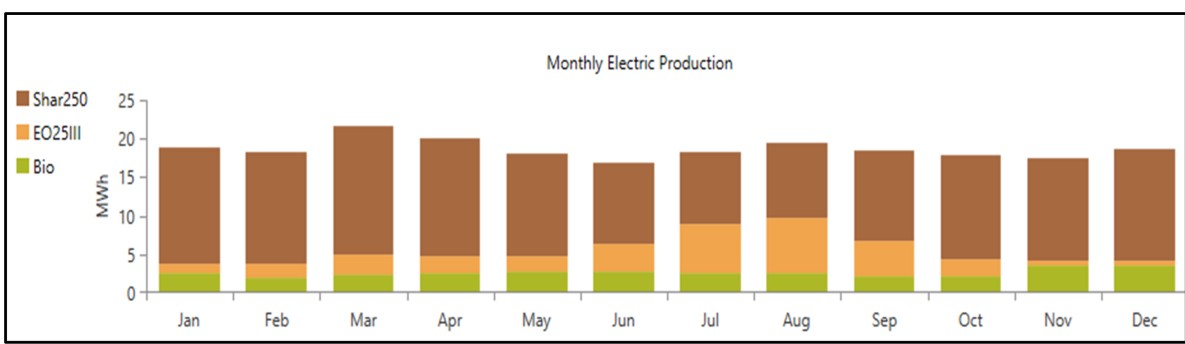

**Figure 6.** Monthly electric production by components.

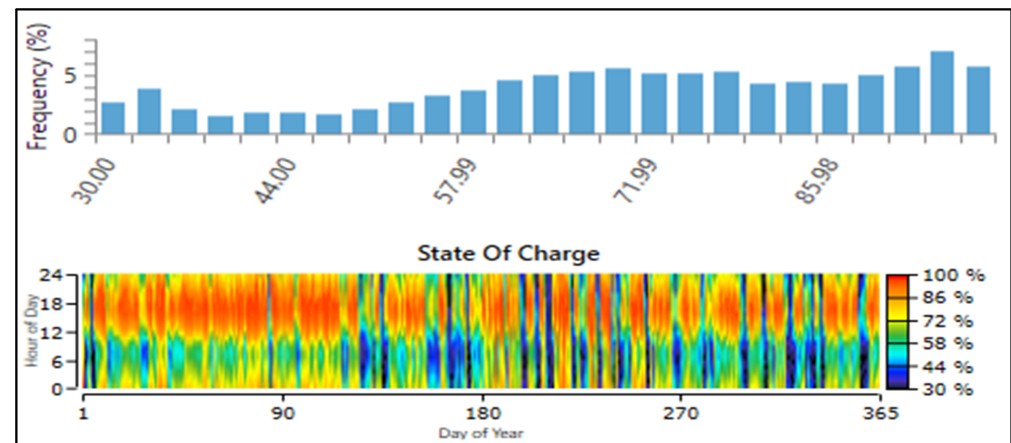

**Figure 7.** Batterystate of charge.

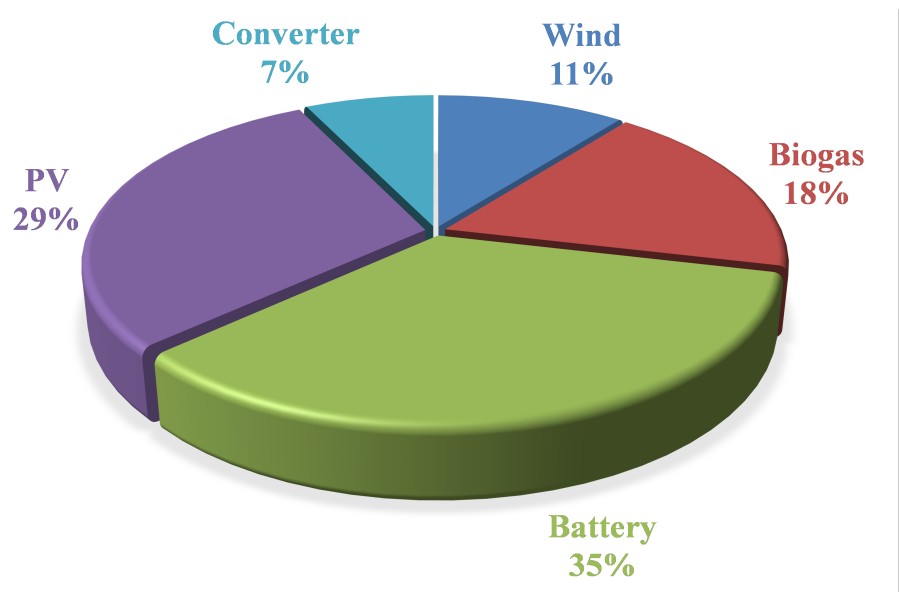

**Figure 8.** Cost analysis.

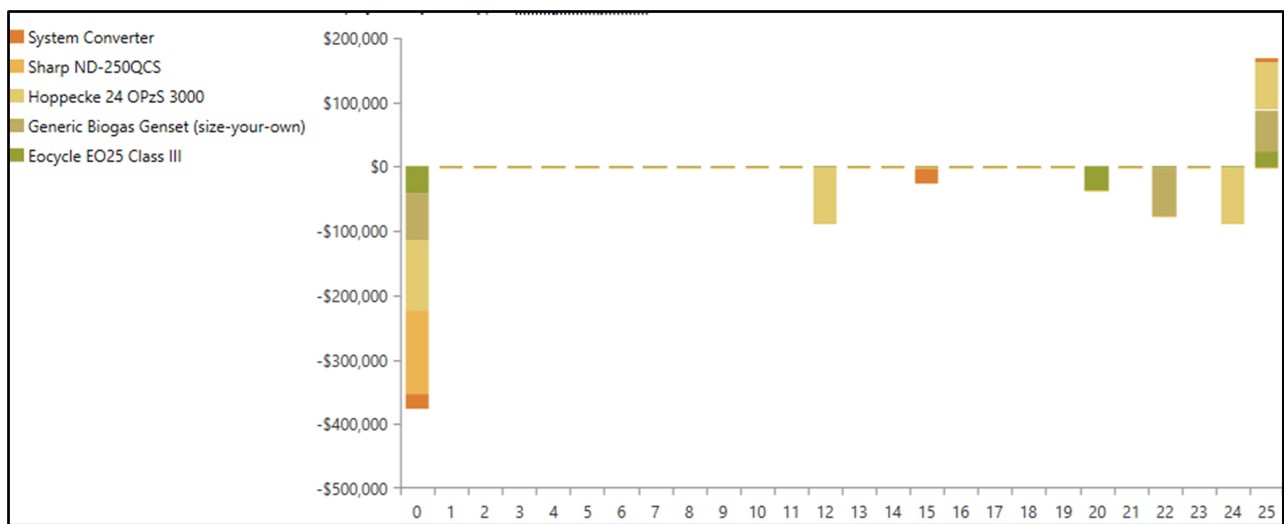

**Figure 9.** Cash flow by components.

**Table 12.** Emission analysis.

| Pollutant (kg/yr) | PV-W-BG-B | DG |
|---|---|---|
| Carbon dioxide | 17.5000 | 152,707 |
| Carbon monoxide | 0.1940 | 790 |
| Unburned hydrocarbons | 0 | 41.9000 |
| Particulate matter | 0 | 6.7600 |
| Sulfur dioxide | 0 | 373 |
| Nitrogen oxides | 0.1210 | 151 |

### 3.3. Sensitivity Analysis on the Optimum System

Sensitivity analysis is usually used to assess the effect of selected parameters on the system's future behavior. The optimum configuration was subjected to a sensitivity analysis in this review, which took into account the discount rate and the battery storage expense. The battery is found to be the most expensive component of the overall system. An analysis is being conducted to determine the effect of rising and decreasing storage costs by 50%. In addition, the discount rate is affected by the country's economic conditions. Sierra Leone's inflation rate is currently unstable. To determine the effect on both the COE and the NPC, the discount rate was increased to 11% and then decreased to 5%. Figure 10 presents the results of the sensitivity analyses of the discount rate with respect to NPC and COE. It can be seen that the discount rate is inversely proportional to the NPC and directly proportional to the COE. Decreasing the interest rate from 8% to 5% increases the NPC from $487,247 to $529,186, which is a 7.9% increment and reduces the COE from $0.211/kWh to $0.169/kWh, which is a 24.85% decrement. This can be verified in Figure 11. In addition, increasing the interest rate to 11% increases the COE by 5.38% and decreases the NPC by 22.5%. A 50% increment in the storage cost of the battery increases both the NPC and COE by 10.26% while a 50% decrement reduces both by 14%. Before a decision is taken, the results of the sensitivity analyses clearly give investors or the government an indication of the effect that improvements in the inflation rate and storage cost would have on the financial output of the project. The Banana islands draw a large number of visitors each year, and government policies on green energy can have a significant effect on the profitability of developing hybrid renewable energy power configurations.

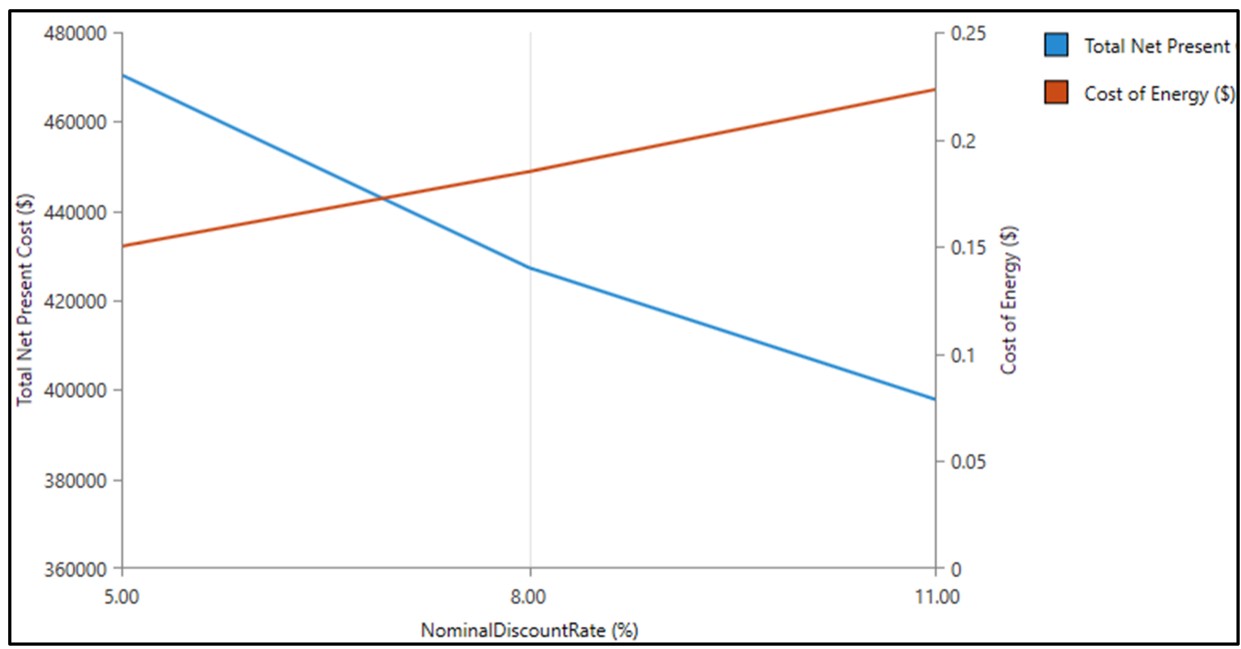

**Figure 10.** Sensitivity analysis: Relation of discount rate to NPC and COE.

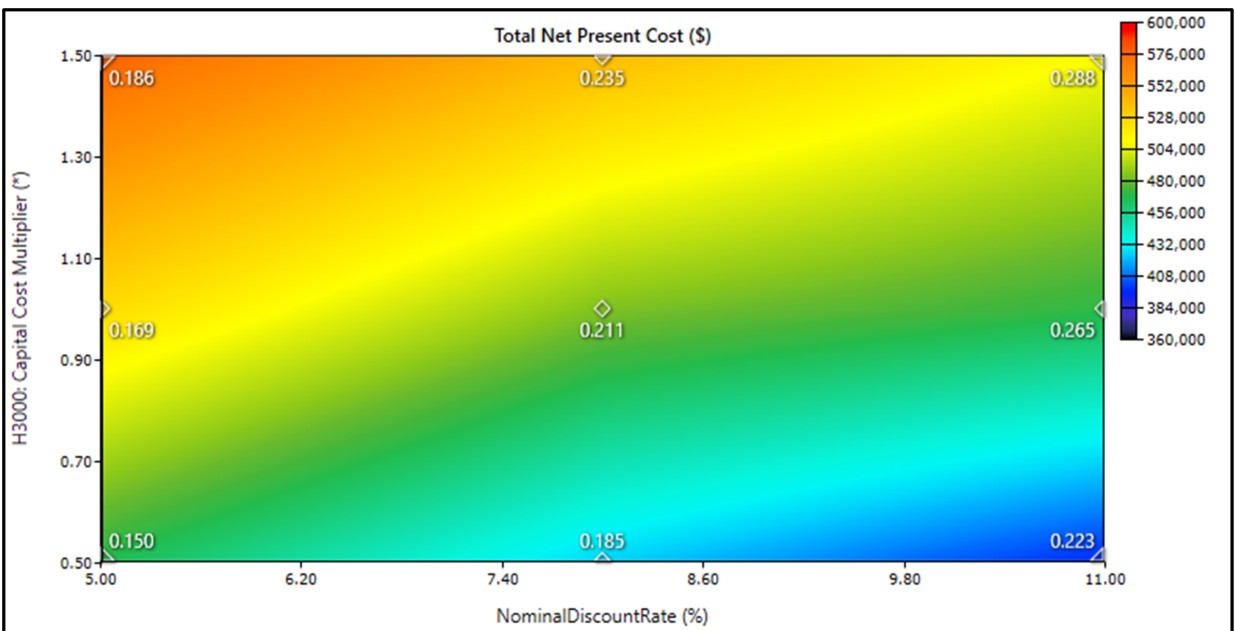

**Figure 11.** Sensitivity analysis: Impact of increasing and decreasing the discount rate and storage cost on NPC and COE.

## 4. Conclusions

In conclusion:

- The results indicate that using the current diesel-based power system to provide continuous electricity for the citizens of Banana Islands is neither cost-effective nor environmentally friendly. This system has a COE of $0.598/kWh and a $CO_2$ emission of 152,707 kg/yr, which are respectively 64.7% and 99.9% higher than the proposed optimum configuration.

- The AHP method of assigning weights is superior to the Entropy method. The optimum configuration selected by the AHP-CODAS approach (PV-Wind-Biogas-battery) is techno-economically superior but environmentally inferior to the configuration selected (Wind-biogas-battery) by the Entropy-CODAS approach.

- The first three ranked configurations using both weight assignment approaches are those that used the biogas generator as backup components. Biomass hybrid configurations tend to be economical and environmentally better than diesel-based backup configurations although technically, diesel-based hybrid configurations are superior. The decision of using either component as backup sources depends on the objectives of the system designer.
- According to renewable energy data, the average wind speed in the region is very low (3.42 m/s), compared to solar radiation of 5.34 kWh/m$^2$/day. The wind speed increases in July and August, while solar radiation is at its lowest in these months. Operating a system based solely on wind or solar energy would not be cost-effective or long-term.
- The effect of fluctuations in the inflation rate and storage cost on the total system cost was confirmed by sensitivity analyses. These criteria, as well as government policies against the establishment of hybrid renewable energy systems, must be considered when making investment decisions.

The following suggestions can be considered for future studies:

- Employing an integrated weight assignment MADM process to select the optimum configuration.
- Undertaking a comparative study between grid-extension to Banana Islands and the use of hybrid renewable off-grid configuration.
- Establishing sensitivity analysis on the load growth and renewable energy resources to know the cost repercussions in the near future.
- Since Banana Islands is a coastal community incorporating, the use of wave energy converters or hydrokinetic turbines could be a future research approach.

**Author Contributions:** Conceptualization, K.V.K.; methodology, K.V.K.; software, K.V.K. and H.T.; validation, K.V.K., H.M. and M.L.O.; formal analysis, K.V.K.; investigation, K.V.K.; resources, K.V.K.; data curation, K.V.K.; writing—original draft preparation, K.V.K. and H.M.; writing—review and editing, K.V.K., N.K. and H.M.; visualization, K.V.K.; supervision, T.S.; project administration, T.S. and K.V.K.; funding acquisition, T.S. All authors have read and agreed to the published version of the manuscript.

**Funding:** This research received no external funding.

**Institutional Review Board Statement:** Not applicable.

**Informed Consent Statement:** Not applicable.

**Data Availability Statement:** Not applicable.

**Conflicts of Interest:** The authors declare no conflict of interest.

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
