# Peer review of "Multi-Attribute Decision-Making Approach for a Cost-Effective and Sustainable Energy System Considering Weight Assignment Analysis"

_sustainability, doi:10.3390/su13105615_

Round 1
Reviewer 1 Report
The authors have presented an interesting study that discussed effective decision making for cost-effective and sustainable energy systems looking critically into its techno-economic assessment which is a great fit for sustainability journal. However, i have alot of issues and concerns with this manuscript in its current form. These issues need to be adequately addressed before this reviewer can recommend the article for publication.
Comments:
- Please re-write the abstract to capture the approach used and most significant findings.
- HOMER, AHP-CODAS were first used in the abstract. Provide their full meaning at its point of first usage in the abstract section.
- The introduction needs more references. Eg statements on seirra leone in line 25-26 needs to be backed by at least a reference.
-
Authors should clearly emphasize the novelty and major contribution of their work at the closing of the introduction.
-
line 40-41 also need some references. Please check throughout the manuscript to provided the needed references. Poor referencing is one of the major issues i have with this study.
-
what is WASPAS? please provide a list of abbreviation with meaning covering all uncommon abbreviations used in this work.
- Line 118 to 129 , page 3, this should be presented at the end of the introduction as the contribution of this study. Its current location looks rather awkward.
- The 'related works" should form a paragraph as a continuation of the introduction, and not a sub-section 1.1 as presented here.
- Highlight the exact location of Banana island in figure 1 using a coloured marker. Reference the map source.
- What does 0-21 stand for on the x axis of the bar chart in Figure 2?
- Figure 4 is confusing and unclear. Revise thee arrows, and include the electric load for the wind source.
- References in Table 1 should appear in the last column (after specification column).
- Authors should be clear on the classification of MADM technique used in their study. Is it pairwise based, scoring based or out ranking based method?
- Line 221, incorrect referencing style. Correct this.
- From line 229, Authors should present the steps (1-9) for implementing the CODAS algorithm using a clear schematic illustration with correct arrows leading to the next step. This will make it easily understood by future readers.
- AHP procedures should also be presented in a stepwise manner using clear diagrams. Line 251
- Inconsistent decimal places in the tables. Check this, adopt a uniform decimal place throughout the tables in the manuscript (2 or 4 dp, not both)
- The way the results of the first approach presented in bullet points does not look good at all. I suggest that the authors present this as a discussion in a very concise manner. See Line 282-308
- Provide some suggestions , perspectives for future studies at the end of your conclusion to ensure the continuity of this interesting area of research.
- Author names in reference list number 1 is incorrect. Fix this .
- On the overall, i strongly advise the authors to thoroughly proof read this manuscript for grammatical and tense errors. From my 1st reading alone, a lot of grammatical errors were detected, alongside some unscientific grammar. Please fix this.
Author Response
Thank you for the review comments. The various changes are highlighted in yellow in the revised manuscript. Please see the attached document for your consideration.

Reviewer 2 Report
The manuscript presented multi-attribute decision making approache to electrify Banana Islands in a cost-effective and sustainable way. Authors have presented the background, methodology, results and discussion in reasonable way. Authors are encouraged to improve the manuscript by considering the following points.
1) Please use a consitence term for the selected location -> banana islands, or Banana island? See page 3-line 116, page 12-line 272, page 4- Fig 1, page 4- 132-134.
2) page 5- line 184- please confirm April is the peak month for solar ration and clearness index. Fig 3 showed March is the peak month. Furthermore, legends for clearness index and solar, Fig 3A are having a same symbols. - change one of them for b/w prints.
3) page 6, line 173 - Check your symbols- No Gs and Kp in equation 1
4) page 6, line 174- please confirm the unit for temperature coefficient . Existing one is wrong.
5) page 7, line 175- please confirm ref [32] is the original source pf eq 2.
6) page 7, line 178 - clarify what is 'ambient temperature at condition 20 dec C'.
7) page 8, equation 4 is not completed.
8) page 8, equation 5 cant be found from ref 34.
9) page 8, check equation 5 - should it be E_sub(load.t) . DA (t) instead of E_sub[(load.t).DA (t)]?
10) page 8, check equation 7.
11) page 8, line 198. explain the meaning of "F1 is the slope = 2 (kg/hr/kW) for a 30-kW generator". where is the slope?
12) page 9, line 103-105; suggest to use a consitence subscript for STP - you used STP and 0.
13) page 9, line 207, what was the interest rate used?
14) page 10, Figure 5, 'decision' symbol is wrongly used for simulation.
15) page 11, equation 13 - is it complete?
16) page 11, line 245, why 0.02 is used as threhold value?
17) page 13, table 3. please discuss results of PV+W+DG+B vs W +DG+B especially O&M cost and CO2.
18) page 14; Fig 6- suggest to move it nearer to the text /discussion at page 16. similarity Tables 4, 5, 6 etc are not presented at the optimal location. For example, you discuss table 4 in subsection 3.2.1. but the nearest table is table 7.
19) page 14; table 7- explanation is required.
20) Tables 7 & 9, Tables 8 & 10 are having same caption and missed important information. Optimisation methods used should to be included in the captions.
21) page 16, line 350. Due to lack of infomation as mentioned in point 20, it is not easy to capture the message in "From results obtained in Table 8 and Table 10, it is apparent the AHP method of assigning ...".
22) improved your list of references. Currently the authors for several papers are wrongly reported- for examples 1, 21, 29, 37 etc.
Lastly authors are encouraged to check all the equations cited.
Thank you,
Author Response

(The authors gave the same response as above.)

Reviewer 3 Report
There is no error within the paper as presented by the authors
Author Response
Thank you for your comments.
Round 2
Reviewer 1 Report
I commend the authors for implementing most of the corrections suggested . Nonetheless, there are still a few minor writing issues that need to be addressed to put this paper in perfect shape before publication.
- The full meaning of HOMER provided in the abstract is wrong. Correct this to "Hybrid Optimization of Multiple Energy Resources". Also correct this in line 114
- Line 18 of the abstract. Format of presenting the NPC and CoE are incorrect. The standard format is $487,247, and $ 0.211/kWh. Please correct to standard throughout the manuscript especially line 249-250, 267-268, 314 among others.
- Full meaning of abbreviations should only be provided once at the point of first usage. Subsequently the abbreviations can be used continuously throughout the text. This is not the case with this study. Please correct this anywhere full meanings have been repeated. See line 114, 214 where full meaning of HOMER is wrongly repeated. Just use HOMER since it has been previously defined.
- Read through again. All Figures and tables must be cited/mentioned in-text before its first appearance. This needs to be corrected.
- Line 29-230 ... first introduced by Saaty? Is Saaty the reference to Saaty [number], or just [number], and list it in the reference list.
- Inconsistent decimal places still exist in Table 3. Authors have 1,2,3 dps all in one table. Be consistent please.
- Tables of Algorithm 1, 2, 3 should be properly named with a title.
Author Response
Thank you for the revision. Please find the attached document for your consideration.
